# Efficacy, Stability, and Safety Evaluation of New Polyphenolic Xanthones Towards Identification of Bioactive Compounds to Fight Skin Photoaging

**DOI:** 10.3390/molecules25122782

**Published:** 2020-06-16

**Authors:** Diana I. S. P. Resende, Mariana C. Almeida, Bruna Maciel, Helena Carmo, José Sousa Lobo, Carlotta Dal Pozzo, Sara M. Cravo, Gonçalo P. Rosa, Aida Kane-Pagès, Maria do Carmo Barreto, Isabel F Almeida, Maria Emília de Sousa, Madalena M. M. Pinto

**Affiliations:** 1Laboratório de Química Orgânica e Farmacêutica, Departamento de Ciências Químicas, Faculdade de Farmácia, Universidade do Porto, 4050-313 Porto, Portugal; dresende@ff.up.pt (D.I.S.P.R.); mariana_c_a@hotmail.com (M.C.A.); scravo@ff.up.pt (S.M.C.); madalena@ff.up.pt (M.M.M.P.); 2CIIMAR-Centro Interdisciplinar de Investigação Marinha e Ambiental, Avenida General Norton de Matos, S/N, 4450-208 Matosinhos, Portugal; 3Laboratório de Tecnologia Farmacêutica, Departamento do Medicamento, Faculdade de Farmácia, Universidade do Porto, 4050-313 Porto, Portugal; brunademaciel@gmail.com; 4UCIBIO, REQUIMTE, Laboratório de Toxicologia, Departamento de Ciências Biológicas, Faculdade de Farmácia, Universidade do Porto, 4050-313 Porto, Portugal; helenacarmo@ff.up.pt; 5UCIBIO/REQUIMTE, MedTec-Laboratório de Tecnologia Farmacêutica, Departamento de Ciências do Medicamento, Faculdade de Farmácia, Universidade do Porto, 4050-313 Porto, Portugal; slobo@ff.up.pt; 6University of Ferrara, Dept. of Life Sciences and Biotechnology, Via Fossato di Mortara 17/1944121 Ferrara, Italy; carlotta.dalpozzo@student.unife.it; 7cE3c–Centre for Ecology, Evolution and Environmental Changes / Azorean Biodiversity Group, 9500-321 Ponta Delgada, Portugal; goncalo.p.rosa@uac.pt (G.P.R.); maria.cr.barreto@uac.pt (M.d.C.B.); 8Faculdade de Ciências e Tecnologia, Universidade dos Açores, 9500-321 Ponta Delgada, Portugal; aidakane.1@hotmail.com

**Keywords:** skin, photoaging, antioxidants, xanthones, 1,2-dihydroxyxanthone, DPPH, elastase, collagenase, tyrosinase, phototoxicity

## Abstract

Antioxidants have long been used in the cosmetic industry to prevent skin photoaging, which is mediated by oxidative stress, making the search for new antioxidant compounds highly desirable in this field. Naturally occurring xanthones are polyphenolic compounds that can be found in microorganisms, fungi, lichens, and some higher plants. This class of polyphenols has a privileged scaffold that grants them several biological activities. We have previously identified simple oxygenated xanthones as promising antioxidants and disclosed as hit, 1,2-dihydroxyxanthone (**1**). Herein, we synthesized and studied the potential of xanthones with different polyoxygenated patterns as skin antiphotoaging ingredients. In the DPPH antioxidant assay, two newly synthesized derivatives showed IC_50_ values in the same range as ascorbic acid. The synthesized xanthones were discovered to be excellent tyrosinase inhibitors and weak to moderate collagenase and elastase inhibitors but no activity was revealed against hyaluronidase. Their metal-chelating effect (FeCl_3_ and CuCl_2_) as well as their stability at different pH values were characterized to understand their potential to be used as future cosmetic active agents. Among the synthesized polyoxygenated xanthones, 1,2-dihydroxyxanthone (**1**) was reinforced as the most promising, exhibiting a dual ability to protect the skin against UV damage by combining antioxidant/metal-chelating properties with UV-filter capacity and revealed to be more stable in the pH range that is close to the pH of the skin. Lastly, the phototoxicity of 1,2-dihydroxyxanthone (**1**) was evaluated in a human keratinocyte cell line and no phototoxicity was observed in the concentration range tested.

## 1. Introduction

Being constantly exposed, the skin is highly affected by external factors like air pollutants and UV radiation. Besides that, food contaminants and toxic drugs can also have damaging effects on the skin [1]. All these aspects, as well as the intrinsic production of reactive oxygen species (ROS), result in oxidative processes that contribute to skin aging as well as to some pathological conditions like skin cancer [2,3]. When the amount of ROS is too high and/or the skin’s intrinsic antioxidant mechanisms are overwhelmed oxidative stress occurs [1]. In order to supplement the organism’s natural defenses, it is important to include antioxidants in the diet. In the case of the skin, there is yet another way to reinforce the cutaneous antioxidant system, which is via the application of topical antioxidant products. It has been proved that the topical application of effective antioxidant formulations can improve several skin parameters and is useful in the reduction of skin aging signs [4,5]. Furthermore, the introduction of antioxidants in sunscreens has shown to promote an increase in photoprotection effectiveness and UV-filter stability [6,7,8].

Repeated exposure to UV radiation leads to the formation of ROS, which triggers a series of processes involved in photoaging [9]. Increased production of elastase, collagenase, and hyaluronidase occurs when UV rays penetrate the dermal layers leading to degradation of collagen, elastin, and hyaluronic acid, respectively. Common consequences are wrinkle formation and sagging of skin, since these macromolecules are responsible for providing strength, elasticity, and moisture to the skin. [10,11]. Additionally, hyperpigmentation occurs due to excessive accumulation of melanin. Tyrosinase is the key enzyme initiating skin pigmentation, and, therefore, the inhibition of this enzyme is one of the most common strategies for reducing pigmentation in the skin [12]. Skin exposition to UV rays also leads to an increase in cutaneous intracellular catalytic ferric levels establishing the potential of ferric chelators for topical treatments [13]. Environmental contaminants, mainly generated by industry and car exhaust, also have a negative impact on the skin, causing premature aging, pigmentation spots, acne, or even more serious dermatological issues. These pollutants include, among others, polycyclic aromatic hydrocarbons, volatile organic compounds, particulate matter, oxides, and heavy metals. They enhance the production of ROS, reducing the content of antioxidants in the skin and causing oxidative stress [11].

Polyphenols have long been recognized as one of the most important groups of antioxidants. Particularly, simple oxygenated xanthones have been reported to have antioxidant activity that varies with the degree of oxygenation as well as with the position of the substituents. This type of naturally occurring xanthones has shown scavenging activity against ROS as well as metal-chelating capacity and inhibition of lipid peroxidation [14,15,16]. By reacting directly with free radicals, they neutralize them and prevent their deleterious effects in the organism. Structure–activity studies suggested that simple xanthones with a catechol unit (vicinal diol groups) have promising antioxidant activity [16,17,18]. The catechol unit is present in other known antioxidants and it had already been linked to the antioxidant activity of other polyphenolic compounds [19]. One possible explanation is that the cooperation between vicinal hydroxyl groups increases the strength of the hydrogen bond that is formed between the xanthone derivative and its target [16]. Additionally, it has been suggested that the radical derived from catechol could be stabilized by the vicinal hydroxyl group and by an intramolecular hydrogen bond [16].

In a previous study developed by our group [16], a library of 20 mono and deoxygenated xanthones was investigated for their antioxidant potential by the 2,2-diphenyl-1-picrylhydrazyl (DPPH) and peroxyl radical scavenging effect as well as the inhibitory effect on the prooxidant enzyme myeloperoxidase (MPO). From the antioxidant screening, 1,2-dihydroxyxanthone (1,2-DHX, **1**, Figure 1) emerged as a hit compound, that was further characterized for its chelating properties and its cytotoxic effect on a human keratinocyte cell line [16]. The results suggested the possible efficacy of xanthone derivatives as antioxidants with potential for skin care application. In order to further explore the potential of 1,2-DHX (**1**), we also investigated its effects on the growth of the A375-C5 melanoma epithelial cell line and its modulatory effects on the activity of the THP-1 macrophage cell line, namely cytokine production. The results suggested a putative impact of 1,2-DHX (**1**) in melanoma treatment, not only due to a direct effect on cancer cells but also by modulation of macrophage activity [20]. Toxicity was excluded by performing the 3-(4,5-dimethylthiazol-2-yl)-2,5-diphenyl tetrazolium bromide (MTT) reduction, neutral red (NR) uptake, Alamar Blue reduction, and Trypan Blue exclusion assays. 1,2-DHX (**1**) did not promote a significant cytotoxic effect up to 50 µM. This compound can be considered noncytotoxic since an 83% decrease in HaCaT cell viability in the neutral red assay was observed. Additionally, 1,2-DHX (**1**) did not show any scavenging activity of NO generated in a cell-free system [16,20].

In the light of these recent findings, the optimization of the xanthone structure aiming the discovery of new antioxidants with potential as skin antiaging actives, led us to assemble a small library of polyoxygenated xanthones with privileged patterns of antioxidant motifs such as catechol (**1**–**3**), hydroquinone (**4**), or *peri-*carbonyl hydroxyl groups (**1**, **4**, and **5**; Figure 1) and to investigate their potential for skin care. Inhibition activities of enzymes related to skin photoaging (tyrosinase, collagenase, elastase, and hyaluronidase), were for the first time investigated for 1,2-DHX (**1**) and related xanthones as well as DPPH reduction activity, metal-chelating effect, and stability at different pH values. For the first time to the best of our knowledge, the phototoxic potential and prooxidant effect of 1,2-DHX (**1**) was also evaluated in a human keratinocyte cell line.

## 2. Results and Discussion

### 2.1. Synthesis

Xanthones **1 [21]**, **2 [22]**, **3 [22]**, and **4 [23]** were prepared as described before with slight modifications (Scheme 1). The synthesis of 1,2-DHX (**1**; Scheme 1a) involved as starting materials methyl 2-bromobenzoate and 3,4-dimethoxyphenol; the resulting methyl 2-(3,4-dimethoxyphenoxy)benzoate, after hydrolysis and subsequent cyclization of 2-(3,4-dimethoxyphenoxy)-benzoic acid with lithium diisopropylamide (LDA) or with AcCl, and further demethylation afforded **1 [21]**. Xanthones **2** and **3** were prepared via a Friedel–Crafts acylation of the appropriate methoxytoluene by 2-methoxybenzoyl chloride, using aluminum chloride as an acid catalyst to give a benzophenone intermediate. Further nucleophilic addition followed by the elimination of methanol under basic conditions and microwave irradiation afforded **2** and **3** (Scheme 1b) [24]. Xanthone **4** was synthesized using a recently reported methodology via acyl radical intermediates starting from salicylaldehyde and 1,2,4-trimethoxybenzene (Scheme 1c) [23]. Coupling of 2-(dimethoxymethyl)phenol (obtained from salicylaldehyde) with 2-bromo-5-methoxycyclohexa-2,5-diene-1,4-dione (obtained from 1,2,4-trimethoxybenzene), cyclization with NBS and benzoyl peroxide and subsequent reduction with zinc and cyclization led to the formation of 1,4-dihydroxy-3-methoxy-9*H*-xanthen-9-one (**4**). The synthesis of a new oxygenated xanthone **5** (Scheme 1d) was performed through a modification to the classic Grover, Shah, and Shah reaction involving the condensation between 2,6-dihydroxy-4-methylbenzoic acid and 5-methylbenzene-1,3-diol [25]. The use of Eaton’s reagent (phosphorus pentoxide and methanesulfonic acid (P_2_O_5_/CH_3_SO_3_H)) as the coupling agent and catalyst instead of a mixture of a phosphorus oxychloride and zinc chloride led to 1,8-dihydroxy-3,6-dimethyl-9*H*-xanthen-9-one (**5**) in 7% yield. To the best of our knowledge, derivatives **1**–**5** are not naturally occurring polyphenols.

### 2.2. Antioxidant Activity

#### 2.2.1. DPPH Reduction Activity

The screening of the reduction activity of xanthones **1**–**5** was firstly estimated by a decolorization assay using DPPH^•^ as free radical (Table 1). Antiradical activity was expressed as the maximum inhibition reached at 25 µM. Among all tested xanthone derivatives, 3,4-dihydroxy-1-methyl-9*H*-xanthen-9-one (**2**) and 3,4,6-trihydroxy-1-methyl-9*H*-xanthen-9-one (**3**) revealed DPPH reduction activity comparable to 1,2-dihydroxy-9*H*-xanthen-9-one (**1**) and with an IC_50_ in the same range of the positive control ascorbic acid. Compound **1** remained as the hit compound with the highest radical scavenging activity [16]. For compounds **4** and **5**, it was not possible to determine the IC_50_ since maximum reduction activity was lower than 50%, probably due to solubility issues.

#### 2.2.2. Metal-Chelating Effect

Interactions between metal ions and organic molecules can easily be detected by UV/Vis. The interaction of the tested xanthones with ferric and cupric ions was evaluated on the basis of changes in the UV/Vis spectra of the free xanthones, following the addition of FeCl_3_ and CuCl_2_, respectively. Based on the DPPH reduction activity, compounds **1**–**4** presented the best results. Although the antioxidant potential of **4** was higher in comparison with **3**, the low solubility of **4** did not allow the determination of the IC_50_. Hence, compounds **1**, **2**, and **3** were selected to be further investigated for their capacity to absorb UV radiation (Figure 2) and their spectra revealed high absorbance in the UVB range (280–320 nm). UVB radiation is responsible for a series of damage effects on the skin, either erythema and burns caused by short-term UV exposure or even photocarcinogenesis after long-term exposure [26]. Sequential additions of FeCl_3_ and CuCl_2_ solutions to 1,2-DHX (**1**) resulted in a significant change of the absorbance spectrum with bathochromic shifts (UV/Vis spectra) from the original band, as previously reported by our group [16]. This shift substantiates the formation of a complex in the presence of Fe (III) and Cu (II) ions due to the presence of a hydroxyl group *peri*-carbonyl emphasizing the potential of 1,2-DHX (**1**) to chelate metals. In order to serve as a comparison, this study was repeated under the same conditions for compounds **2** and **3**. On the other hand, for 3,4-dihydroxy-1-methyl-9*H*-xanthen-9-one (**2**) and 3,4,6-trihydroxy-1-methyl-9*H*-xanthen-9-one (**3**) bathochromic shifts were observed only with sequential additions of FeCl_3_ while no effect was observed when CuCl_2_ was added. This fact validates the importance of a *peri*-carbonyl hydroxyl group in xanthone **1**. Considering the preliminary results regarding the antioxidant activity of the selected xanthones, we can infer that xanthone **1** exhibits a dual ability to protect the skin against UV damage by combining antioxidant/metal-chelating properties with UV-filter capacity.

### 2.3. Inhibition of Elastase, Collagenase, Hyaluronidase and Tyrosinase Activity

The inhibitory activities of xanthones **1**–**3** against enzymes related to skin photoaging were screened in vitro by colorimetric methods. Each enzyme was incubated with xanthones **1**–**3** and after the addition of the specific substrates, its activity was measured and compared to the activity of a control without the xanthone derivative. The enzymes screened were tyrosinase, elastase, collagenase, and hyaluronidase. and the results obtained are summarized in Table 2. The IC_50_ of compounds **1–3** in each activity was calculated when the % of activity inhibition at 150 µM was higher than 50%.

All the xanthone derivatives tested, 1,2-DHX (**1**), 3,4-dihydroxy-1-methyl-9*H*-xanthen-9-one (**2**), and 3,4,6-trihydroxy-1-methyl-9*H*-xanthen-9-one (**3**), were found to be extremely good tyrosinase inhibitors, with IC_50_ in the same order of magnitude and lower than the one obtained for the positive control kojic acid. Besides reports of other xanthone derivatives, like 3-aryl substituted xanthones [27] or pyranocycloartobiloxanthone A [28], being very active against tyrosinase, this is, as far as we were able to confirm, the first time that the tyrosinase inhibition activity of simple oxygenated xanthones such as compounds **1**–**3** is being reported.

Contrary to the observed for tyrosinase inhibition, compounds **1**–**3** presented low activity against elastase. Compound **1** exhibited the strongest effect by inhibiting 35.2% of elastase activity at the maximum concentration tested (150 µM). For the collagenase inhibition assay, all three compounds presented weak to moderate activity, inhibiting the enzymatic activity by 25–35% at 150 µM. None of the compounds had inhibitory activity against hyaluronidase.

### 2.4. Stability

Considering the results obtained in the previous screening assays and their feasible synthesis and potential to scale-up, xanthones **1** (1,2-DHX) and **2** were selected to proceed with additional tests with the final purpose to be used in a skin care formulation. Since the quality of the active ingredient can change with time under the influence of various environmental factors, it is a primary concern to the formulator to assess physical and chemical stability as it affects the safety and efficacy of the cosmetic product [29,30] To establish the stability of xanthones **1** (1,2-DHX) and **2,** their behavior versus time was observed in pre-established special conditions that accelerate the degradation of compounds. Hence, **1** (1,2-DHX) and **2** were submitted under a range of pH buffers to test their pH stability for 21 days. As far as the efficacy of the skin care formulations is concerned, the pH is regarded as a significant parameter. The pH of human skin normally ranges from 4.5 to 6.0. Therefore, in order for a formulation present the highest skin compatibility, it should have a pH that is close to this range [3,31,32]

The graphs depicting absorbance versus time (0, 1, 2, 24, 192, 360, and 504 h) at different pH are overlapped for xanthones **1** (1,2-DHX) and **2** (Figure 3A,B, respectively) in order to understand the variation in the absorbance peak. Considering xanthone **2**, a degradation of the compound for the range of pH tested overtime is observed. 1,2-DHX (**1**) revealed to be more stable in the pH range that is closest to the pH of the skin, thus presenting a potential to be used as a future cosmetic active agent.

### 2.5. Phototoxicity

Current regulatory guidelines demand preclinical photosafety evaluation within the pharmaceutical and cosmetic industry framework. According to the animal testing banning put in force by EU Regulation 1223/2009, only in vitro tests can be performed for cosmetic ingredients. Phototoxicity was herein evaluated in a human keratinocyte cell line, using a method adapted from the OECD 432 guideline and previously implemented in our lab (HaCaT, neutral red uptake assay (NRU)) [33]. This provides a more realistic experimental model in comparison with the 3T3 NRU assay preconized in the OECD guideline, performed with mice fibroblasts [34].

1,2-DHX (**1**) was not cytotoxic to HaCaT cells even after irradiation. The IC_50_ values and consequently the photo-irritation factor (PIF) could not be obtained. However, as the cell viability was not decreased after UV exposure (Figure 4), the compound is deemed nonphototoxic up to 200 µM. 1,2-DHX (**1**) did not promote any significant increase in the fluorescence signal of the dichloro-dihydro-fluorescein diacetate (DCFH-DA) probe, relative to solvent control, supporting the absence of ROS generation upon incubation with HaCaT cells (Figure 5).

In the same way as other catechol containing polyphenols [35], 1,2-DHX (**1**) did not generate prooxidant effects, which are usually associated with the presence of one phenol group on the aromatic ring, since the derived phenoxyl radicals tend to be very unstable.

## 3. Materials and Methods

### 3.1. Materials

All reagents and solvents were purchased from TCI (Tokyo Chemical Industry Co. Ltd., Chuo-ku, Tokyo, Japan), Acros (Geel, Belgium), Sigma-Aldrich (Sigma-Aldrich Co. Ltd., Gillingham, UK), or Alfa Aesar (Thermo Fisher GmbH, Kandel, Germany) and had no further purification process. Solvents were evaporated using a rotary evaporator under reduced pressure, Buchi Waterchath B-480 (BÜCHI Labortechnik AG, Switzerland). The reaction was monitored by TLC carried out on precoated plates with 0.2 mm thickness using Merck (Darmstadt, Germany) silica gel 60 (GF254) with appropriate mobile phases and detection at 254 and/or 365 nm. Purification of the synthesized compound was performed by chromatography flash column using Merck silica gel 60 (0.040–0.063 mm). Melting point (mp) was measured in a Köfler microscope (Wagner and Munz, Munich, Germany) and is uncorrected. ^1^H and ^13^C-NMR spectra were taken in CDCl_3_ at room temperature on a Bruker Avance 300 (Bruker Biosciences Corporation, Billerica, MA, USA) instrument (300.13 or 500.13 MHz for ^1^H and 75.47 or 125.77 MHz for 13C). Chemical shifts are expressed in δ (ppm) values relative to tetramethylsilane (TMS) as an internal reference. Coupling constants are reported in hertz (Hz). ^13^C-NMR assignments were made by 2D HSQC and HMBC experiments (long-range C, H coupling constants were optimized to 7 and 1 Hz). HRMS mass spectra were measured on a Bruker Daltonics micrOTOF Mass Spectrometer, recorded as ESI (electrospray) mode in Centro de Apoio Científico e Tecnolóxico á Investigation (C.A.C.T.I.), University of Vigo, Spain. Immortalized human keratinocyte (HaCaT) cell line was obtained from Cell Lines Service (CLS, Eppelheim, Germany).

Tyrosinase, L-tyrosine, kojic acid, monosodium phosphate, sodium phosphate dibasic elastase, *N*-methoxysuccinyl-Ala-Ala-Pro-Val-*p*-nitroanilide, *N*-methoxysuccinyl-Ala-Ala-Pro-chloro, 2-[4-(2-hydroxyethyl)piperazin-1-yl]ethanesulfonic acid (HEPES), *N*-[3-(2-furyl)acryloyl]-Leu-Gly-Pro-Ala (FALGPA), *N*-[tris(hydroxymethyl)methyl]-2-aminoethanesulfonic acid (TES) sodium salt, ninhydrin, citric acid, sodium citrate, EDTA, hyaluronidase, hyaluronic acid, calcium chloride, sodium aurothiomalate, potassium metaborate, 4-dimethylaminobenzaldehyde (DMAB), sodium acetate, acetic acid, and hydrochloric acid were purchased from Sigma-Aldrich. Collagenase was supplied by Merck. Tin (II) chloride was obtained from Riedel-deHaën (Seelze, Germany). The water used in all experiments was purified water obtained using a Direct-Q Water Purification System (Merck Millipore, Burlington, Massachusetts, United States) with a reverse osmosis process.

### 3.2. Synthesis of 1,4-dihydroxy-3-methoxy-9H-xanthen-9-one (**5**)

A mixture of phosphorus pentoxide (1.418 g, 5.000 mmol) and methanesulfonic acid (30 mL) was heated at 90 °C until a clear solution was obtained (30 min). Then, a mixture of 5-methylresorcinol (0.517 g, 4.162 mmol) and 2,6-dihydroxy-4-methylbenzoic acid (0.700 g, 4.162 mmol) was added to the reaction mixture. This mixture was stirred at reflux for 1 h and the progress of the reaction was monitored by TLC. Then, the resulting mixture was slowly poured into crushed ice and allowed to stand overnight in the fridge. The solid was collected by filtration, washed with water, and dried. Purification by column chromatography using a mixture of EtOAc/hexane (1:9) as mobile phase gave 1,4-dihydroxy-3-methoxy-9*H*-xanthen-9-one (**5**) in 7% yield. 1,4-Dihydroxy-3-methoxy-9*H*-xanthen-9-one (**5**); White solid (72.7 mg, 7% yield); m.p. 171–173 °C. 1H-NMR (CDCl_3_, 300.13 MHz): δ = 8.18 (d, ^3^*J*_8,7_ = 8.8 Hz, 1H, H-8), 6.94 (dd, ^3^*J*_7,8_ = 8.8 Hz, ^4^*J*_7,5_ = 2.4 Hz, 1H, H-7), 7.52 (d, ^4^*J*_5,7_ = 2.4 Hz, 1H, H-5), 4.06 (s, 3H, 3-OCH_3_), 4.03 (3H, s, 4-OCH_3_), 3.94 (3H, s, 6-OCH_3_), 3.06 (s, 3H, 1-CH_3_) ppm. ^13^C-NMR (CDCl_3_, 75.47 MHz): δ = 176.9 (C-9), 164.8 (C-6), 156.6 (C-10a), 153.8 (C-3), 151.4 (C-4a), 139.7 (C-4), 136.8 (C-1), 134.6 (C-6), 128.5 (C-8), 118.0 (C-9a), 117.7 (8a), 116.5 (C-2), 113.6 (C-7), 99.7 (C-5), 61.9 (3-OCH_3_), 61.2 (4-OCH_3_), 55.9 (6-OCH_3_), 21.5 (1-CH_3_) ppm. HRMS (ESI^+^): m/z [C_17_H_15_BrO_5_ + H]^+^ calcd. for [C_17_H_16_BrO_5_]: 379.0176; found 379.0161.

### 3.3. DPPH Reduction Assay

The reduction of the stable DPPH free radical was measured by monitoring the change in absorbance at 517 nm. These experiments were made in a 96-well microplate. Compounds **1, 2** and **3** were dissolved in ethanol 70%, and compounds **4** and **5** were dissolved in a solution of ethanol 70% with 1% DMSO. In each experiment, the compounds were tested in the concentration range 0.7812 μM to 50 μM, in triplicate. For each sample concentration, a blank with ethanol 70%, instead of DPPH was made. Ascorbic acid was used as a positive control and tested in the same conditions. Blanks containing only ethanol 70% and control containing DPPH in ethanol 70% (190 μM) were used. The absorbance at 517 nm was measured in a microplate reader BioTek^TM^ Synergy HT (BioTeck Instruments, Vermont, USA) after 60 min of incubation in the dark with a temperature-controlled holder at 25 °C. The microplate was shaken before each reading.

The scavenging percentage of DPPH was calculated using the following equation:(1)$$\%\mathrm{scavenging}\;\mathrm{of}\;\mathrm{DPPH}=100\times \left(1-\frac{{\mathrm{Abs}}_{\mathrm{sample}+\mathrm{DPPH}}-{\mathrm{Abs}}_{\mathrm{sample}\;\mathrm{blank}}}{{\mathrm{Abs}}_{\mathrm{DPPH}}-{\mathrm{Abs}}_{\mathrm{EtOH}}}\right)$$

The IC_50_ value for each sample and for the ascorbic acid was obtained by linear regression.

### 3.4. Metal-Chelating Effect

The ultraviolet/visible (UV/Vis) spectra were recorded in methanol using a path length of 1 cm, at room temperature on a Varian CARY100 spectrophotometer (Agilent Technologies, Santa Clara, California, United States) from a range of 200 nm to 500 nm (software CaryWin UV, v. 3.0). The metal-chelating effect was evaluated by described procedures [36] with stock solutions (1.0 × 10^−4^ M) in methanol. Iron (III) and copper (II) solutions were prepared in a concentration of 1.0 × 10^−3^ M from FeCl_3_ and CuCl_2_ in deionized water. Briefly, sequential additions of 33 μL of metal ion solution were performed to 3 mL of compounds **1**, **2**, and **3**.

### 3.5. Inhibitory Activities of Enzyme Related with Skin Aging

#### 3.5.1. Tyrosinase Inhibition Assay

This activity was assayed by an adaptation of the tyrosinase inhibition method described by Shimizu et al. [37] and modified by Manosroi et al. [38] Briefly, 25 μL of tyrosinase enzyme solution (135 U/mL), 25 μL of ten serial concentrations of the compounds (0.3–150 μM dissolved in 0.1 M phosphate buffer pH 6.8 containing no more than 2.5% DMSO) and 100 μL phosphate buffer were mixed in a 96-well plate and incubated at 37 ± 2 °C for 20 min. Then, 50 μL of 1.66 mM of tyrosine solution in 0.1 M phosphate buffer (pH 6.8) were added. The enzyme activity was measured at 490 nm every 10 min for 30 min in a Bio-Rad Model 680 Microplate Reader (Bio-Rad Laboratories, Inc., Hercules, CA, USA). Kojic acid was used as positive control. The experiments were done in triplicate. For each concentration, enzyme activity was calculated as a percentage of the velocities compared to that of the assay using buffer without any inhibitor. The IC_50_ value was determined as the concentration of a compound that inhibited 50% of enzyme activity.

#### 3.5.2. Elastase Inhibition Assay

The compounds were assayed by the method described by Ndlovu et al. [39] with some modifications. Briefly, 25 μL of elastase enzyme solution (0.3 U/mL), 50 μL of ten serial concentrations of the compounds (0.3–150 μM dissolved in 0.1 M HEPES buffer pH 7.5 containing no more than 2.5% DMSO) and 125 μL HEPES buffer were mixed in a 96-well plate and incubated at room temperature for 20 min. Then, 50 μL of *N*-Methoxysuccinyl-Ala-Ala-Pro-Val-*p*-nitroanilide (1 mM) were added. The enzyme activity was measured at 405 nm in the moment of substrate addition and after 40 min of incubation at 25 °C in a Bio-Rad Model 680 Microplate Reader (Bio-Rad Laboratories, Inc., Hercules, CA, USA). *N*-Methoxysuccinyl-Ala-Ala-Pro-Chloro was used as positive control. The experiments were done in triplicate. For each concentration, enzyme activity was calculated as a percentage of the velocities compared to that of the assay using buffer without any inhibitor. The IC_50_ value which was the concentration of the sample that inhibited 50% of the enzyme activity was determined.

#### 3.5.3. Collagenase Inhibition Assay

An adaptation of the method of Moore and Stein [40] with modifications by Mandl et al. [41] was used to determine anticollagenase activity. To 2 mL test tubes was added: 25 μL of collagenase solution (0.8 U/mL), 25 μL TES buffer (50 mM) with 0.36 mM calcium chloride (pH 7.4) and 50 μL of test sample or the reference compound EDTA (with concentrations ranging between 9.4 and 150 μM). The tubes were incubated in a water bath at 37 °C for 20 min. Thereafter, 50 μL FALGPA (1 mM) solution was added to the tubes and incubated further for 60 min at 37 °C. To all tubes, 200 μL of a solution containing equal volumes of a 1.6 mg/mLTin chloride (II) solution in 200 mM citrate buffer (pH 5) and 50 mg/mL ninhydrin solution in DMSO was added. All tubes were placed in a water bath (100 °C) for 5 min and left to cool to room temperature before adding 200 μL of 50% isopropanol to each tube. Contents in the tubes were then transferred to respective wells in 96-well plates. Absorbance was detected at 550 nm in a Bio-Rad Model 680 Microplate Reader (Bio-Rad Laboratories, Inc., Hercules, CA, USA). Percentage of collagenase inhibition was calculated as:(2)$$\%\;\mathrm{collagenase}\;\mathrm{inhibition}\;=\frac{{\mathrm{Abs}}_{\mathrm{control}}-{\mathrm{Abs}}_{\mathrm{sample}}}{{\mathrm{Abs}}_{\mathrm{control}}}\times 100$$
where Abs_control_ is the absorbance of buffer + collagenase; Abs_sample_ is the absorbance of buffer + collagenase + sample/standard. All assays were carried out in triplicate and results expressed as % of collagenase inhibition and/or IC_50_, i.e., as the concentration yielding 50% of collagenase inhibition, calculated by interpolation from the % collagenase inhibition vs concentration curve.

#### 3.5.4. Hyaluronidase Inhibition Assay

The methods described by Ndlovu et al. [39] and Zhou et al. [42] were applied for the hyaluronidase inhibition assay. Into 2 mL test tubes was placed: 50 μL of calcium chloride (12.5 mM), 50 μL of test samples or sodium aurothiomalate diluted in 0.1 M acetate buffer; pH 3.5 (with concentrations ranging between 9.4 and 150 μM), and 25 μL hyaluronidase (0.5 mg/mL). The tubes were incubated in a water bath (37 °C; 20 min) after which 50 μL of the substrate hyaluronic acid (0.25 mg/mL in 0.1 M acetate buffer; pH 3.5) was added and the tubes incubated for further 40 min. Twenty-five microliters of KBO_2_ • 4 H_2_O (0.8 M) was added to all tubes which were placed in a water bath (100 °C) for 3 min, left to cool to room temperature, and 800 μL of DMAB (4 g DMAB in 40 mL acetic acid and 5 mL 10 N HCl) was added. The tubes were then incubated for 20 min and the contents transferred to respective wells in a 96-well plate. Absorbance was detected at 585 nm in a Bio-Rad Model 680 Microplate Reader (Bio-Rad Laboratories, Inc., Hercules, CA, USA). Percentage of hyaluronidase inhibition was calculated as:(3)$$\%\;\mathrm{hyaluronidase}\;\mathrm{inhibition}\;=\frac{{\mathrm{Abs}}_{\mathrm{control}}-{\mathrm{Abs}}_{\mathrm{sample}}}{{\mathrm{Abs}}_{\mathrm{control}}}\times 100$$
where Abs_control_ is the absorbance of buffer + hyaluronidase; Abs_sample_ is the absorbance of buffer + hyaluronidase + sample/standard. All assays were carried out in triplicate and results expressed as % of hyaluronidase inhibition and/or IC_50_, i.e., as the concentration yielding 50% of hyaluronidase inhibition, calculated by interpolation from the % hyaluronidase inhibition vs concentration curve.

### 3.6. Stability

Buffer solutions at pH values of 2, 3, 4, 5, 6, 7, and 8 were prepared and the pH of each buffer was measured with a pH meter. All the solutions were prepared following Portuguese Pharmacopeia 9.0 specifications (F.P. (VIII) 2008).

A solution at 10 mM in DMSO of each compound was prepared. One hundred microliters of antioxidant solution were diluted to 5 mL of the different buffered pH solutions, prepared in triplicate. Their absorbance measured by UV/Vis spectrophotometry in the wavelength range between 200 and 500 nm at time intervals of 0, 1, and 2 h and 1, 8, 15, and 21 d at room temperature. Each pH buffer solution was used as a compensation solution.

### 3.7. Safety Evaluation

#### 3.7.1. Cell Culture

The HaCaT cells were maintained at 37 °C in a humidified atmosphere of 95% air and 5% CO_2_ in the incubator in dulbecco’s modified eagle’s medium (DMEM) with 10% fetal bovine serum (FBS) and 1% antibiotics. Using an inverted microscope, cell confluence was observed and if the cells reached 70–80% confluence, subculture was done to prevent cell death. For this purpose, the culture medium was aspirated and the cells were washed with dulbecco’s phosphate buffered saline (DPBS), 2 mL of trypsin-EDTA were added and incubated for 5–8 min at 37 °C in a 5% CO_2_ atmosphere. After cell detaching, medium was added in order to block the trypsin action and cell suspension was centrifuged at 416 g for 5 min. The supernatant was discarded, the pellet suspended in culture medium, and cells were counted. For cell counting: 50 μL of cell suspension were added to 50 μL of Trypan Blue vital dye (1:10), resuspended, placed in a Neubauer chamber (Hirschmann, Germany) and the viable cells were counted. For cell freezing, DMSO (5% *v*/*v*) was used as a cryopreservative to prevent the formation of crystals during the storage phase.

#### 3.7.2. ROS Generation Assay

Cells (2 × 10^4^ cells/well) were seeded on 96-well tissue culture plates and incubated at 37 °C in a 5% CO_2_ atmosphere for 24 h. Afterwards, the medium was removed, cells were washed with DPBS, and complete DMEM containing 50 μM DCFH-DA was added and incubated for 30 min. Afterwards, the cells were washed once with DPBS and 1,2-DHX (**1**; 12.5, 25, 50, 100, and 200 µM) was added and incubated for 24 h. H_2_O_2_ (700 µM) was used as positive control, and untreated cells were taken as the negative control. After incubation, the fluorescence was measured using a fluorescence plate reader with a baseline of 485 nm excitation and 530 nm emission. The study of the putative interference of compound and solvent fluorescence was previously performed. The results were calculated as the ratio between the fluorescence of treated cells and untreated cells and expressed in percentage.

#### 3.7.3. Phototoxicity Evaluation

HaCaT cells were seeded onto 96-well tissue culture plates (Orange Scientific, Braine-l’Alleud, Belgium) at a 2 × 10^4^ cells/well density and incubated at 37 °C in a 5% CO_2_ atmosphere for 24 h. Afterwards, the medium was removed, cells were washed with DPBS, and fresh DMEM without FBS and phenol red containing the different concentrations of 1,2-DHX (**1**; 12.5, 25, 50, 100, and 200 µM) was added and incubated under the same conditions for 1 h. For each experiment, two identical plates were seeded and incubated with the test compound. One plate was kept in the dark while the other plate was irradiated for 10 min [OSRAM Ultra Vitalux 240V E27 lamp (Munich, Germany), UVA irradiation dose of 1.7 mW/cm^2^] with the temperature kept at 29–32 °C. The cells were then washed once again with DPBS and the medium was replaced with fresh DMEM without phenol red and incubated for 18–22 h. After this incubation period, cells from both plates were washed with DPBS and complete DMEM containing 50 μg/mL NR was added to each well and incubated for 3 h. Since the NR could precipitate, NR solutions were prepared every second day and incubated overnight at 37 °C in a 5% CO_2_ atmosphere protected from light. Before the addition to the wells, the NR solution was centrifuged at 1500 g for 10 min and filtered (5 μm). After this, the NR solution was removed, cells were washed once with DPBS previously heated at 37 °C, and an NR desorb solution (50% ethanol/1% acetic acid/49% distilled water) was added to extract the NR dye from the cells. For the reading procedure, each plate was placed in a microplate shaker (BioTek, United States) for 10 min, at room temperature and protected from light, to extract all the NR from the cells and obtain a homogeneous solution. In this last step, the absorbance was measured at 540 nm. Within each plate, adequate solvent controls were tested. Cell viability data obtained from each plate was expressed as the absorbance ratio of treated to solvent control cells and was further used to estimate the IC_50_ values using linear regression analysis. The PIF is the ratio between the IC_50_ value of the irradiated (Irr+) versus the IC_50_ value of the nonirradiated (Irr-) cells. According to the OECD 432 guideline, a PIF index < 2 predicts the absence of a phototoxic effect, a PIF index between 2 and 5 predicts a probable phototoxic effect, and a PIF > 5 predicts a phototoxic effect [34].

### 3.8. Statistical Analysis

The assays were carried out in triplicate. The results were expressed as mean values ± standard deviations. The data was analyzed using a one-way ANOVA with Tukey’s test; *p* < 0.05 was considered statistically significant.

## 4. Conclusions

A series of five polyoxygenated xanthones were synthesized and their antioxidant activity was evaluated by the DPPH assay. Further studies on their capacity to inhibit skin-degrading enzymes, metal-chelating effect as well as the stability at different pH values led to one hit compound, 1,2-DHX (**1**). Results of these studies indicated that 1,2-DHX (**1**) displays a dual ability to protect the skin against UV damage by combining antioxidant/metal-chelating properties with UV-filter capacity, and revealed to be more stable in the pH range that is closest to the pH of the skin, with no phototoxicity observed in the concentration range tested. This study reinforces the safety and efficacy of a dihydroxyxanthone as a potential antiaging compound paving the way to in vivo studies.

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
