# Peer review of "Efficacy, Stability, and Safety Evaluation of New Polyphenolic Xanthones Towards Identification of Bioactive Compounds to Fight Skin Photoaging"

_molecules, 2020, doi:10.3390/molecules25122782_

Round 1

Reviewer 1 Report

Figure 2 UV.Vis spectra of compounds with addition of FeCl3 and CuCl2 are confusing and their presentation should be reconsidered- a purpose of the arrow “sequential additions, bathochromic shifts” is not clear. One representative example of the effect showing the spectra in conjunction with a table listing the wavelengths of maxima and corresponding concentrations of Fe3+ and Cu2+ ions might be considered to replace Figure 2. In this context, a description of the effect in lines 159-161, 5/16 is relatively vague. [What may be a reason for the appearance of sharp valleys at ~350 nm in shown spectra?]

3/16, line 100 “did not promoted

Many acronyms are not defined – e.g. 3/16 MTT; 12,13/16 FBS; 12,13/16 DMEM; 12,13/16 DPBS-

3.1. Materials do not include the acquisition method for HaCaT cells

3.6. Citations for the preparation of buffers is appropriate and sufficient information, there is no need for detailed recipes

Author Response

Figure 2 UV.Vis spectra of compounds with addition of FeCl3 and CuCl2 are confusing and their presentation should be reconsidered- a purpose of the arrow “sequential additions, bathochromic shifts” is not clear. One representative example of the effect showing the spectra in conjunction with a table listing the wavelengths of maxima and corresponding concentrations of Fe3+ and Cu2+ ions might be considered to replace Figure 2. In this context, a description of the effect in lines 159-161, 5/16 is relatively vague. [What may be a reason for the appearance of sharp valleys at ~350 nm in shown spectra?]

The valuable suggestions from reviewer 1 were added to Figure 2. The effect is due to the transition between tungsten and deuterium lamps that, in this specific assay, is not attenuated by baseline correction due to the concentration of the compounds.

 3/16, line 100 “did not promoted

Sentence was changed accordingly to “did not promote”

Many acronyms are not defined – e.g. 3/16 MTT; 12,13/16 FBS; 12,13/16 DMEM; 12,13/16 DPBS-

Acronyms descriptions were added accordingly.

3.1. Materials do not include the acquisition method for HaCaT cells

Details regarding acquisitions of the cell line are now described in the manuscript.

3.6. Citations for the preparation of buffers is appropriate and sufficient information, there is no need for detailed recipes

Buffers preparation description was removed from the text.

Reviewer 2 Report

The manuscript by Resende et al. is focus on study of antioxidant and anti-photoaging activity of xanthones as well as the analysis of their stability and safety. Xanthones belong to polyphenols and can be obtained both via isolation (e.g. from plants) or chemical synthesis. In manuscript Authors synthesized and analyzed 5 xanthones differed in their structure. The measured parameters were: antioxidant potential (all compounds), metal chelation and enzyme inhibition potential (3 compounds from 5) and phototoxicity (only X1 i.e. 1,2-DHX). Also the stability of 2 compounds under the influence different pH and times has been checked. Manuscript is interesting, good written and describes an important results.  The article can be published after minor corrections.

Remarks:

  1. I recommend place the information about synthesis of xanthon 5 (lines 118-123) after the synthesis description of other compounds (lines 124-136).
  2. It should be also marked in column description “DPPH reduction activity (%) at 25µM” that reduction activity was also measure at 60 min.
  3. Why Authors did not take also Xanthon 4 for analysis of metal chelation. If choice was connected only with DPPH reduction activity then the antioxidant potential of X4 was higher in comparison with X3 (34.6 vs. 24.9 respectively).
  4. In line 164 Authors wrote: “this study was repeated under the same conditions for compounds 1-3.”, but in lines 159-163 the chelation potential of X1 has already been demonstrated. Does it mean that once again the analysis was repeated for X1. Maybe should be wrote that for compounds 2-3. Please clarify it.
  5. Below the table 2 (lines 188-189) was wrote e.g. Kojic acid (tyrosinase). Such description implicate that tyrosinase is the synonym of Kojic acid. Better is write e.g. Kojic acid (for tyrosinase) MAAPVCK (for elastase) etc. were used as positive control.
  6. Why also DMSO was used in preparation of X4 and X5 solutions (lines 293-294) ?
  7. What kind of solvent was used to prepare FeCl3 and CuCl2 solutions?
  8. In line 171 Authors wrote: “Inhibitory activities of elastase, collagenase, hyaluronidase and tyrosinase”. This subsection title implicate that enzymes were analyze as inhibitors. The title should be improved in order to clearly demonstrate that xanthones were analyzed as the enzymes inhibitors.
  9. Authors wrote in Materials and methods (lines 350-352) that results were expressed as IC50. Then why the IC50 values for collagenase are not present in table 2.
  10. In lines 367-369 Authors informed about hyaluronidase inhibition, calculated by interpolation from the % collagenase inhibition vs concentration curve. How hyaluronidase inhibition can be calculated via interpolation from the % collagenase inhibition?
  11. In lines 445-447 Authors wrote, that the PIF was calculate. On the other hand in lines 231-232 was wrote that the PIF could not be obtained. Please explain it.
  12. In lines: 337 and 354 for references numbers the superscripts should be removing.
  13. In lines: 414 and 426 (where Authors give the information about number of cells) the superscript for 4 should be done.
  14. Some abbreviations in the manuscript, i.e.: NRU, NR, PIF do not have the explanation.

Author Response

  1. I recommend place the information about synthesis of xanthon 5 (lines 118-123) after the synthesis description of other compounds (lines 124-136).

Text regarding the synthesis of xanthone 5 was moved to appear after the synthesis description of compounds 1-4.

  1. It should be also marked in column description “DPPH reduction activity (%) at 25µM” that reduction activity was also measure at 60 min.

Sentence was changed accordingly to “DPPH reduction activity (%) at 25 µM (at 60 min)”.

  1. Why Authors did not take also Xanthon 4 for analysis of metal chelation. If choice was connected only with DPPH reduction activity then the antioxidant potential of X4 was higher in comparison with X3 (34.6 vs. 24.9 respectively).

We understand the reviewer concern; to clarify this aspect the following sentence was added to the revised manuscript  “Based on the DPPH reduction activity, compounds 1-4 presented the best results. Although the antioxidant potential of 4 was higher in comparison with 3, the low solubility of 4 didn’t allow the determination of the IC50.”

  1. In line 164 Authors wrote: “this study was repeated under the same conditions for compounds 1-3.”, but in lines 159-163 the chelation potential of X1 has already been demonstrated. Does it mean that once again the analysis was repeated for X1. Maybe should be wrote that for compounds 2-3. Please clarify it.

We clarify this aspect by relacing 1-3 by 2 and 3 in the revised manuscript.

  1. Below the table 2 (lines 188-189) was wrote e.g. Kojic acid (tyrosinase). Such description implicate that tyrosinase is the synonym of Kojic acid. Better is write e.g. Kojic acid (for tyrosinase) MAAPVCK (for elastase) etc. were used as positive control.

Sentence was changed to “Kojic acid (for tyrosinase), MAAPVCK (for elastase), and EDTA (for collagenase) were used as positive controls.”.

  1. Why also DMSO was used in preparation of X4 and X5 solutions (lines 293-294) ?

To clarify this aspect, the following sentence was added in the legend of table 1 in revised manuscript: “*Compounds 1, 2 and 3 were dissolved in ethanol 70 %. Due to low solubility in ethanol 70%, compounds 4 and 5 were dissolved in a solution of ethanol 70 % with 1% DMSO.”

  1. What kind of solvent was used to prepare FeCl3 and CuCl2 solutions?

Solvent (deionized water) was added to the text (experimental section).

  1. In line 171 Authors wrote: “Inhibitory activities of elastase, collagenase, hyaluronidase and tyrosinase”. This subsection title implicate that enzymes were analyze as inhibitors. The title should be improved in order to clearly demonstrate that xanthones were analyzed as the enzymes inhibitors.

Subsection title was changed into “Inhibition of elastase, collagenase, hyaluronidase and tyrosinase activity”

  1. Authors wrote in Materials and methods (lines 350-352) that results were expressed as IC50. Then why the IC50 values for collagenase are not present in table 2.

Sentence “The IC50 of compounds 1-3 in each activity was calculated when the % of activity inhibition at 150 µM was higher than 50%.” Was added to the text in section 2.3.

Sentence “ (…) results expressed as % of collagenase inhibition and/or IC50…” was added in material and methods section

  1. In lines 367-369 Authors informed about hyaluronidase inhibition, calculated by interpolation from the % collagenase inhibition vs concentration curve. How hyaluronidase inhibition can be calculated via interpolation from the % collagenase inhibition?

There was an error in the text. % collagenase was now replaced by % hyaluronidase.

  1. In lines 445-447 Authors wrote, that the PIF was calculate. On the other hand in lines 231-232 was wrote that the PIF could not be obtained. Please explain it.

The sentence in lines 445-447 was changed in order to describe only the meaning of PIF. Although the purpose was to calculate PIF,  1,2-DHX was not cytotoxic  in the concentration range tested and thus this index could not be calculated.

  1. In lines: 337 and 354 for references numbers the superscripts should be removing.

Superscripts in reference numbers were removed.

  1. In lines: 414 and 426 (where Authors give the information about number of cells) the superscript for 4 should be done.

Superscript was done in both numbers.

  1. Some abbreviations in the manuscript, i.e.: NRU, NR, PIF do not have the explanation.

Abbreviations descriptions were added to the text accordingly.

Reviewer 3 Report

The manuscript aims to evaluate the synthesis of some xanthones and their activity in as skin anti-photoaging ingredients.

The manuscript is sound and data are reliable. I have two major issues:

- There is a missing of statistical analysis of the data. First of all a paragraph in mat met section is required to explain the analysis. Then, in all the figures and tables there is a missing of letters/asterisk to explain the differences.

- Can authors provide information whether the synthesized compounds are available in some foods or they are just chemically synthesized for this paper? Being polyphenols a class of compounds mainly introduced by diet, I am wondering whether those compounds might be introduced through the diet and maybe exert the bioactivity also after digestion.

Author Response

There is a missing of statistical analysis of the data. First of all a paragraph in mat met section is required to explain the analysis. Then, in all the figures and tables there is a missing of letters/asterisk to explain the differences.

A section regarding statistical analyses was included in the manuscript. Superscript letters were added to the tables. The footnote “In each column, different letters indicate significant differences (p < 0.05).” was added to the tables.

Can authors provide information whether the synthesized compounds are available in some foods or they are just chemically synthesized for this paper? Being polyphenols a class of compounds mainly introduced by diet, I am wondering whether those compounds might be introduced through the diet and maybe exert the bioactivity also after digestion.

The synthesized compounds are obtained chemically and, to the best of our knowledge, are not available in foods. Nevertheless, closed related analogues like 2-hydroxy-1-methoxyxanthone is from natural occurrence. To clarify this aspect, the following sentence was added to the revised manuscript: “To the best of our knowledge, derivatives 1-5 are not naturally-occurring polyphenols.”. Reviewer 3 idea is interesting, however further studies on the toxicity of these compounds are needed towards this use.

Round 2

Reviewer 3 Report

I have no further comments on the paper.
Just add the name of the software used for the statistical analysis.